# Representation with a capital 'R': measuring functional alignment with causal perturbation

**Jacob S. Prince**
Department of Psychology
Harvard University
Cambridge, MA 02138
jacob.samuel.prince@gmail.com

**George A. Alvarez**
Department of Psychology
Harvard University
Cambridge, MA 02138
alvarez@wjh.harvard.edu

**Talia Konkle**
Department of Psychology
Center for Brain Science
Kempner Institute
Harvard University
Cambridge, MA 02138
talia_konkle@harvard.edu

## Abstract

Ambiguity surrounding the term 'representation' in biological and artificial neural systems hampers our ability to assess their alignment. In this paper, we draw a critical distinction between two notions of representation: the conventional 'representation' as a mere encoding, and 'Representation-with-a capital-R', which entails functional use within a system. We argue that while current methods in neuroscience and artificial intelligence often focus on the former, advancing our understanding requires a shift toward the latter. We critique existing linking methods such as representational similarity analysis and encoding models, highlighting key limitations in their ability to capture functional correspondence. We then propose an updated paradigm that involves specifying explicit models of information readout and testing their functional properties using causal perturbation. This framework, which treats neural networks as a new species of model organism, may help reveal the principles governing functional representations in both biological and artificial systems.

## Introduction

The goal of understanding how the visual system represents and transforms information has driven widespread efforts to model brain responses using deep encoding models. Initiatives like Brain-Score (1, 2), Algonauts (3), and Sensorium (4) benchmark the predictive power of deep neural networks (DNNs) against large-scale datasets of neural activity and behavior, aiming to reveal core principles of information processing. These efforts and other related studies (e.g. 5) have sparked extensive debate about the most appropriate methods for encoding: linear vs. nonlinear regression models (6), rotation-invariant vs. rotation-sensitive mappings (7, 8), and, approaches that emphasize tuning properties vs. the overall geometry of the representations (9, 10, see 11 for review). Why, as a field, have we yet to reach consensus on which procedures provide the most valuable insights? This lack of consensus may stem from an over-reliance on representational measures that fail to capture the functional aspects of the underlying processes. At the heart of this debate is a deeper issue:

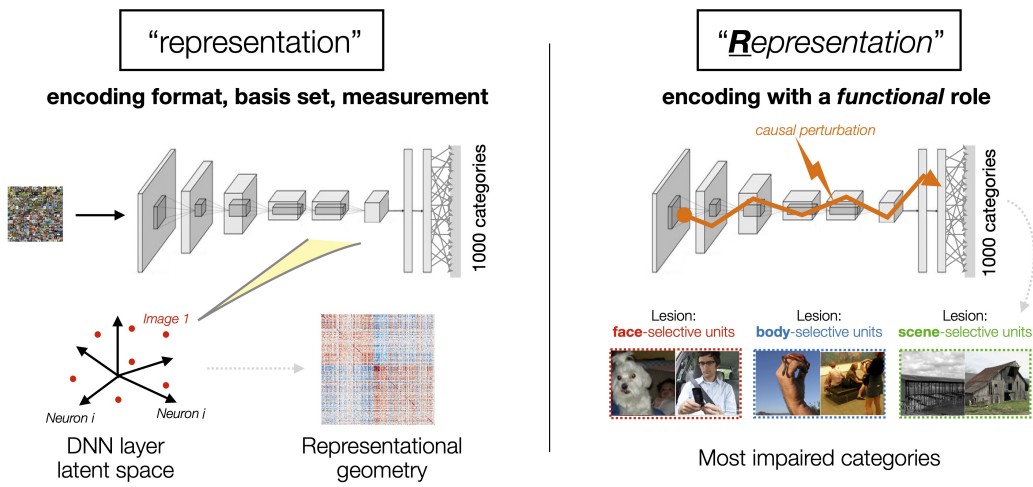

Figure 1: *Two notions of what we mean by "representation."*

viewing 'representation' merely as an encoding format fails to account for the functional role these representations play in downstream processes.

## Two competing notions of representation

Encoding models and representational similarity analysis (RSA, 12) index what we call **representation with a lowercase 'r'**: how information is encoded in a system. This is a broad notion of representation, reflecting any static mapping of stimuli into different bases, such as neural measurement units or model activations, or even pixel space. The modeling aim is to capture the structure of the encoding, and understand its format. This approach is safe, pragmatic, leveraging measurable data from biological and artificial systems and allowing for ranking of DNN feature spaces based on predictivity. However, typical similarity metrics stop far short of addressing whether a given set of internal representations play a similar functional role within both systems. As a result, they provide an incomplete picture of how encoded information is actually used to drive behavior or cognition.

In contrast, **Representation with a capital 'R'** refers to the functional use of encoded information within the system (**Figure 1**). It is not enough for a model to account for the brain's encoding format of stimuli; for information to qualify as capital 'R' Representation, the system must actively use it in ways that mirror the biological processes driving perception or behavior. This idea, rooted in philosophical discussions of mental representation (see 13 for review), emphasizes the functional role of different activity patterns with respect to behavior (see also 14, 15, 16, 17). For example, patterns of activity across face-selective neurons can reliably enable experimenters to decode non-face object information such as houses, chairs, and other objects (18). However, if this encoding does not contribute functionally to the brain's recognition of these items (19, 20), it would fall short of capital 'R' Representation. The key is functional relevance: the system must **read out** the content meaningfully. Testing for these signatures requires causal perturbation of the system, for instance by silencing model units to simulate the effect of lesions on behavior.

## Why do we need Representation with a capital 'R'?

### Lower-case 'r' representations capture a static, often disconnected, view of neural activity

While vision models increasingly succeed in matching neural data from different stages of the visual hierarchy in isolation, they often fail to replicate the actual interactions governing information flow from V1 to high-level visual cortex. The best layer for V1 in one model might appear mid-way through, while for V4, it might come from an earlier layer in another model. Feature reweighting makes divergence from hierarchical correspondence even more likely, and there's often no expectation that one single model will successfully account for multiple stages of the hierarchy (though some

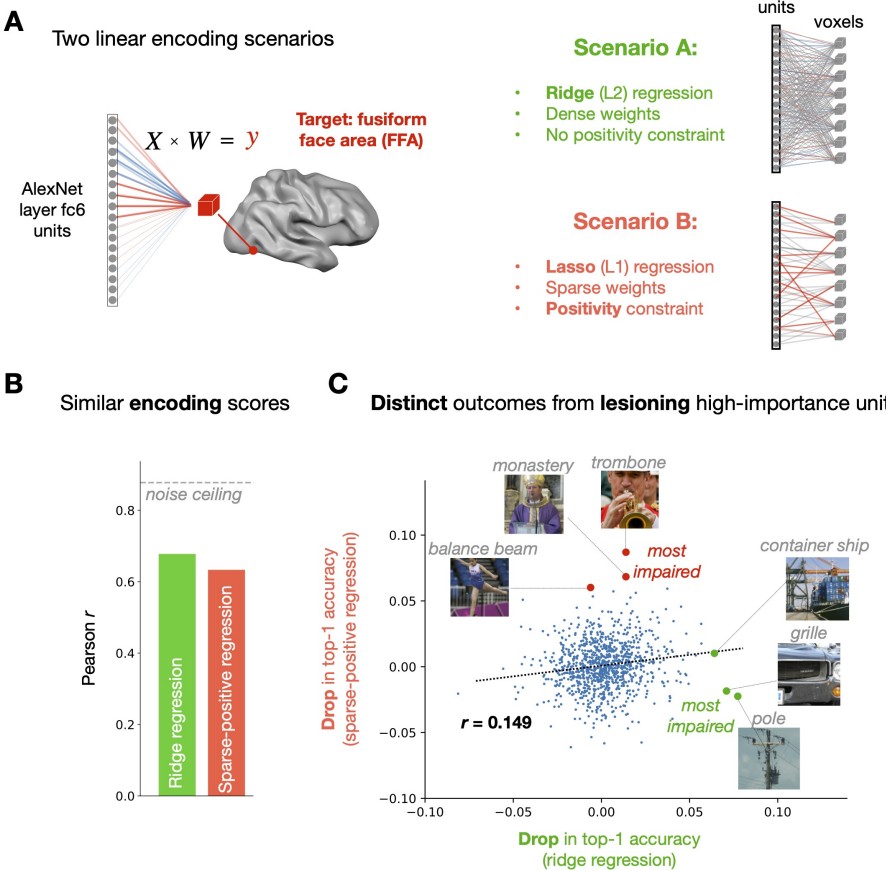

Figure 2: ***Dissociating representational and functional alignment.*** *(A) Two different linear encoding models are fit to predict mean responses in the fusiform face area (FFA) from AlexNet layer fc6. Data are from the Natural Scenes Dataset (25), subject 01, right-hemisphere. Scenario A uses ridge (L2) regression with no positivity constraint, while Scenario B uses lasso (L1) regression with a positivity constraint. (B) Pearson correlation scores when predicting mean FFA responses to a held-out validation set of 1000 images. (C) Functional impact of lesioning the fc6 units with top 1% greatest absolute encoding weight magnitude. The scatter plot shows the drop in top-1 accuracy for different ImageNet categories when lesioning the critical units for each scenario.*

have pursued these modes of comparison, e.g. 21, 22, 23). This limitation is evident in the current implementation of Brain-Score, where any layer of any model can be deemed the best match for a given benchmark. The risk with this approach is that it may isolate models based on superficial pattern matching, without necessarily advancing our understanding of the causal principles driving visual function.

### Representation with a capital 'R' entails causal perturbation of neural systems

We advocate for developing new alignment procedures that directly test whether the information measured in a system is functionally used. This is important, as current linking methods may convey that different models have similar alignment, while masking that they make very different functional predictions. To illustrate this point, we implement a simple case study involving two linear encoding models of brain responses from the fusiform face area (FFA) of human visual cortex (**Figure 2A**). The two regression scenarios both take as input activations from the same DNN layer (AlexNet fc6), but map onto FFA data using different regularization constraints (see *Appendix* for complete details). One approach uses standard ridge regression, while the other employs sparse (lasso) regression with an added positivity constraint (20, 24).

Both regression methods yield comparable predictions on held-out data, appearing similar on the surface in their ability to account for lower-case 'r' representations. However, with free access to the models' internals, we can perform causal perturbations to test the underlying commitments of these different encoding models. After lesioning the 1% of layer units that received the highest absolute weight (importance) in the regression, we observe that the two models produce very distinct patterns of functional predictions about what impairments are likely. For example, across the 1000 ImageNet categories, we computed which categories would have the most impaired recognition, finding a correlation between the two models' lesion-induced deficits of only $r=0.149$. Moreover, several of the most-impaired categories for lesions derived from sparse-positive regression contained faces (e.g. trombone, balance beam), unlike those derived from ridge regression weights (e.g. container ship, pole).

In sum, while encoding models may show similar performance in a lowercase 'r' sense of representation, they can obscure critical (and untested) differences in a capital 'R' sense. Causal perturbations, such as lesioning highly weighted units, provide a powerful way to distinguish models' functional alignment and reveal the different commitments they make about underlying processes. Models that predict equally well under normal conditions can behave very differently when the systems are perturbed, providing key insights about the relationship between activation patterns and functional properties. These signatures may support stronger metrics of alignment to human brain and behavior.

## Toward mapping methods that respect functional properties

Overcoming these limitations requires making *explicit* commitments about how information is used and propagated within both systems. Popular linking methods like RSA and encoding models make *implicit* assumptions about readout. For instance, RSA assumes that tuning directions in neural populations do not matter for readout, as arbitrary rotations of the feature space do not alter the overall geometry. Instead, RSA focuses on matching population-level geometry between model layers and brain areas, assuming that the downstream system has access to the entire population of neural signals. Similarly, encoding models allow for arbitrary transformations of model unit tuning relative to their brain counterparts, implying that tuning matters less in models than it does in the brain. To move beyond these limitations, it is essential to make explicit commitments about how information flows through the system (e.g. 26) and how representations are read out to drive behavior.

In feedforward hierarchical ReLU models, tuning directions—the patterns of a unit's responses to different inputs—are critical for information propagation. In these models, only positive signals are passed on to subsequent layers, making tuning a key determinant of function. Neuropsychological evidence suggests a similar link between high neural activity (i.e., selectivity) and functional outcomes in the brain's visual system (27, 28, 29, 30). By designing comparison methods that respect these known signatures, we may naturally gain deeper insight into functional roles of representations.

In line with this, some have proposed incorporating positivity and sparsity constraints into linear encoding models of the ventral visual stream to pressure them toward better alignment with capital 'R' representations (24). The positivity constraint minimizes tuning reorientation (how much the feature weights warp the original tuning directions of the model layer), which reflects the operating principles of networks that propagate only positive activations. The sparsity constraint reduces feature remixing, encouraging a one-to-one alignment between DNN and brain feature tuning (see also 10). We hypothesize that these constraints will lead to model-to-brain links that better capture the functional relevance of information flow in both systems, providing a more faithful index of capital 'R' representation.

## Challenges in measuring capital 'R' representations

What techniques will promote better investigation of functional or mechanistic alignment? Lesion studies, transcranial magnetic stimulation (31, 32, 33), electrical microstimulation (34), pharmacological inactivation (35), and optogenetics (36) are powerful tools for understanding causal structure-function relationships, but they are costly and difficult to conduct. This hinders our ability to directly observe how representations in one brain region contribute to downstream functions.

DNNs enable us to simulate scenarios that are difficult or impossible to explore in biological systems. By systematically lesioning or pruning layers, or inactivating specific subcircuitry, we can study

how representations in earlier layers contribute to those in downstream layers, and directly link each to functional outcomes (37, 38). This provides a powerful framework for exploring capital 'R' representations. Complementary insight could be gained from applying recent techniques from the fields of mechanistic interpretability and AI safety, which explore how information routes and circuits operate in neural networks (e.g. 39, 40, 41, 42).

In this way, we stand to learn about information transmission and Representation with a capital 'R' by studying the models directly. Each DNN can be conceived of as a simplified model organism, with more constrained inputs and outputs relative to the human brain. Crucially, a model's ability to perform meaningful tasks, such as object categorization, makes it valuable to study in its own right, regardless of whether its solution is human-like. The key idea is that the more we understand the internal processes of these models, and how different computational units contribute to the behavior of interest, the more likely we are to uncover principles of information transmission that could shed light on the function of biological systems.

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

## Appendix

**Supplementary encoding methods for FFA analysis**

The encoding analyses involved data from right-hemisphere FFA-1 from subject 01 of the Natural Scenes Dataset (25). The subject-native surface space data preparation was used, and a stringent noise ceiling signal-to-noise ratio (NCSNR) threshold of 0.4 was applied prior to analysis. For the sake of simplicity, the average activation within the FFA-1 region was used as the encoding prediction target.

We fit linear encoding models from the basis of AlexNet (43) layer fc6 activations. The default ImageNet-pretrained (44) version of AlexNet included in the TorchVision library was used. We implemented two encoding scenarios: for Scenario A, we used unconstrained ridge regression, and for Scenario B, we used lasso regression with a positivity constraint implemented via the sklearn 'Positive=True' argument. Optimal ridge and lasso alpha hyperparameter values were obtained by iteratively fitting the models using a training set of 1000 images, testing either a range of 25 log-spaced values between $10^{-2}$ and $10^8$ for ridge, or, 25 log-spaced values between $10^{-3}$ and $10^{0.2}$ for lasso. Each model was scored using an independent validation set of 1000 images, and the models with the best-scoring alpha values for ridge and positive-lasso were used for subsequent lesioning analyses.

Encoding scores were computed as the Pearson correlation between the model-predicted mean responses of the region and the true mean responses of the region. A noise ceiling value for the performance of computational models of the mean FFA-1 response profile was computed by inputting the matrix of shape (1 neuroid, 1000 images, 3 trials) to the same analytical formula that was used to compute NCSNR for individual voxels, as described in 25.

To assess the functional role of units that received high importance in the regression fits, we first applied a threshold to identify the top 1% of fc6 units with the highest *absolute* weight value, separately for the ridge and positive-lasso scenarios. This meant that 41 units out of the full 4096 in the layer were identified for each lesion. Prior to implementing the lesions, we assessed baseline top-1 ImageNet category recognition over the 1000 categories using the validation stimuli. Then, separately for the two regression scenarios, and with the entire model frozen, we implemented a lesion to the fc6 layer by setting the outputs of the critical units to 0. At this point, with no further re-training of the model, we again assessed category-wise ImageNet recognition performance and computed the category-wise lesioning cost as the drop in top-1 accuracy caused by each of the two lesions. The scatter plot in **Figure 2C** compares the 1000-dimensional category cost profiles obtained from these two lesions. For the sake of visualization, a small amount of random jitter (values drawn from a normal distribution with mean 0, standard deviation 0.01) was added to each datapoint's x and y coordinates. The reported Pearson similarity value between the cost profiles does not consider this jitter.

