# OpenReview forum: "Representation with a capital 'R': measuring functional alignment with causal perturbation"
_NeurIPS.cc/2024/Workshop/UniReps — UniReps_

### Official Review · Reviewer_UfuL · 2024-10-04
**Will spark useful discussions**

**Rating:** 7
**Confidence:** 4

**Review:**

This is a well-written and thought-provoking short abstract. The authors argue that we as a community should place greater emphasis on causal perturbations and understanding of deep networks and biological systems, rather than just characterizing representational geometry (devoid of any context). It serves as sort of a call-to-action and poses more questions than it provides answers. I think it would be a good conversation starter at the workshop.

The one tidbit of analysis shown in Figure 1C is interesting, but I was confused on a critical detail. Did the authors re-train the linear regression models after lesioning the units in the deep network in order of importance? Or was the linear regression model frozen as well? To me, it would make sense to re-fit the regression model after you lesion a unit because the LASSO regularization will tend to zero out units with redundant information. Thus, if you lesion the only remaining unit with a certain kind of tuning (assuming LASSO lesioned out units with correlated tuning) you'd see a giant drop in performance. However, in this scenario re-fitting the LASSO regression from scratch would result in a less significant drop in performance.

I think the author's result is probably interesting either way, but the details should be clarified.

---

### Official Review · Reviewer_aTRa · 2024-10-05
**Review of Submission67**

**Rating:** 4
**Confidence:** 3

**Review:**

This paper raises some criticisms to the current approach to assessing similarity between representations, especially between artificial and biological neural networks. The authors argue that current representational similarity analysis, focused on population-level statistical properties of representations, falls short of detecting true functional similarity between data encodings. Instead, they propose to focus on a different concept of Representation, that takes into account the functional role that encodings have in subsequent read out layers.

Strengths:

- The paper is well-written and easy to follow.
- The idea that statistical similarity between representations is not enough to draw parallels between very different systems is reasonable and well justified.

Weaknesses:

- My main concern with this paper is that, while the authors raise relevant criticisms against the current approach to representational similarity, they do not really propose any actionable way to put their insight into practise.

---

### Official Review · Reviewer_VUqg · 2024-10-06
**Compelling case with interesting direction, but could better address representation analysis literature**

**Rating:** 7
**Confidence:** 4

**Review:**

**Summary**:
- This perspectives paper concretizes an often-discussed distinction between basic encodings (lowercase representations) and representations with functional roles (uppercase 'R' Representations). The authors argue that current neuroscience and AI methods too-often focus on the former, while advancing our understanding requires a shift toward the latter. They critique existing methods like representational similarity analysis and propose a new paradigm based on encoding methods that respect the functional properties of representations.

**Strengths**:
1. Thoughtfully discusses the notion of how representations can be more meaningfully understood in the context of the downstream behaviors they cause and functional relevance
2. Effectively points out gaps in what traditional representation analysis methods (e.g. RSA) offers the Neuroscience and AI communities in terms of understanding
3. The causal perturbation approach to illustrating how encoding methods differ is quite clever

**Weaknesses**:
1. There are existing ways to study representations in a functionally-relevant and task-relevant way. The paper correctly addresses RSA but does not discuss existing alternative methods to study representations in more functionally-relevant ways. For example, [Cohen et al (2020)](https://doi.org/10.1038/s41467-020-14578-5).

---

### Official Review · Reviewer_TEJx · 2024-10-06
**An Interesting Position Paper**

**Rating:** 7
**Confidence:** 5

**Review:**

- Summary:
   - This paper explores the idea that many (all?) methods for comparing representations suffer from a common flaw: they do not privilege information content in representations that is more *functionally relevant* in their comparisons. For example two methods may produce high similarity scores (as measured by linear regression, RSA variants, or global shape metrics) by being similar only in terms of features that are discarded in subsequent stages of computation.
   - The author's suggest that in order to measure the functional similarity between different representations, we ideally need to conduct causal perturbations (i.e. simulating lesions) and observe the impact on downstream behavioral outputs. They conduct one such experiment to disambiguate between two candidate mapping methods between an artificial representation (AlexNet) and fMRI responses in the fusiform face area (FFA). They find that linear mappings that use (1) ridge regression and (2) Lasso regression with a positivity constraint on the weights produce similar levels of predictivity, but distinct predictions when units identified as important for neural prediction are lesioned. Namely, lesioning the Lasso-important units  seems to produce more behavioral impairment for image classes containing faces than lesioning Ridge-important units.

- Strengths:
   - Clarity: ideas experiments are clearly presented, and the analysis of results is both fair and reasonably supports the arguments made in the text.
  - Originality: while a slew of works have considered both novel methods for representational comparisons and the implications of choosing one method over another, the approach of using "artificial" lesioning as a simple way to attempt to measure the functional similarity of how information is encoded in distinct representations, is to my knowledge novel (and potentially very interesting).
  - Fair accounting of weaknesses: the difficulty in conducting lesion studies in-vivo with current experimental methodologies prevents this method of "functional comparison" from closing the loop (i.e. determining whether lesioning FFA produced similar behavioral deficits as lesioning Lasso-important units in Alex-Net). This limitation is clearly addressed in the main text and highlighting this as a weakness is important and valuable to the community (as it helps to dream up solutions!).

- Weaknesses/Suggestions:
  - On the positivity constraint: this is a bit of a nit, but I am not sure that this is necessarily a very large issue within current common practices in terms of linear regression from model to brain. For example many models simply use the outputs of ReLU layers to map to brain responses, so there is no issue with inverting these activities from the functional perspective (i.e. negative weight connections to downstream layers could be functionally important). The point about inverting pre-nonlinearity activations (that will be thresholded to zero) is well taken, but a quick survey of models on the BrainScore platform shows that many models already use the outputs of the nonlinearity to perform the mapping (see for example the top-of-the-leaderboard model here: https://www.brain-score.org/model/vision/646)
  - On the sparsity of mappings (More of a question/thought than a weakness): Should how many model units should be mapped to a specific type of brain measurement depends on the supposed level of physiological correspondence between the representations? For example, if we believe model units are akin to single cell responses than striving for a 1-to-1 mapping between units and Ephys single unit measurements seems reasonable, but would also imply that a large number of model units should be weighted/pooled in order to explain, say, an fMRI voxel response. In general I think it would be interesting to consider or comment on how the issues outlined in the paper relate to/are impacted by the fact that neural responses are measured with a wide variety of resolutions.
  - Measuring functional similarity between pairs of Artificial networks: I think repeating the artificial lesioning experiments but instead on model-to-model comparisons (i.e. where AlexNet is mapped to ResNet via Ridge/Lasso regression) would be both interesting in its own right and a step toward more directly demonstrating that sparse regressions better capture functional similarity, even though these kind of two-way lesioning experiments are currently mostly impractical in the brain-to-model setting.

---

### Decision · Program_Chairs · 2024-10-10

**Decision:**

Accept (Oral)

**Comment:**

In light of the positive reviewers' feedback and relevancy of the submission, we are pleased to accept this paper for presentation at UniReps 2024. We kindly ask the authors to incorporate the reviewers' suggestions and feedback in the final camera-ready version of the manuscript.